# Exponentially Complex “Classically Entangled” States in Arrays of One-Dimensional Nonlinear Elastic Waveguides

**DOI:** 10.3390/ma12213553

**Published:** 2019-10-29

**Authors:** P.A. Deymier, K. Runge, M. A. Hasan, L. Calderin

**Affiliations:** Department of Materials Science and Engineering, The University of Arizona, Tucson, AZ 85721, USA; krunge@email.arizona.edu (K.R.); mdhasan@email.arizona.edu (M.A.H.); lcalderin@email.arizona.edu (L.C.)

**Keywords:** elastic waveguides, classical entanglement, nonlinear elasticity

## Abstract

We demonstrate theoretically, using multiple-time-scale perturbation theory, the existence of nonseparable superpositions of elastic waves in an externally driven elastic system composed of three one-dimensional elastic wave guides coupled via nonlinear forces. The nonseparable states span a Hilbert space with exponential complexity. The amplitudes appearing in the nonseparable superposition of elastic states are complex quantities dependent on the frequency of the external driver. By tuning these complex amplitudes, we can navigate the state’s Hilbert space. This nonlinear elastic system is analogous to a two-partite two-level quantum system.

## 1. Introduction

The notions of superposition of states and entanglement lay at the core of today’s second quantum revolution [1]. Nonlocality and nonseparability are two distinctive attributes of entangled superpositions of states. While nonlocality is a unique feature of the strangeness of quantum mechanics, nonseparability is not. The notion of classical “entanglement”, i.e., local nonseparable superposition of states has received a lot of attention from the theoretical and experimental point of views in the field of optics [2,3,4]. Recently, remarkable new behaviors of sound, analogous to quantum physics, have also been revealed [5]. For instance, elastic waves in one-dimensional (1D) waveguides with broken time reversal or parity symmetry have been shown to obey Dirac-like equations and possess spin-like topology [6,7]. The amplitude of these pseudospin elastic waves takes the form of a spinor in the two-dimensional Hilbert space of the direction of propagation along the waveguides. In parallel arrays of elastically coupled one-dimensional waveguides, the amplitude also spans an N dimensional Hilbert subspace, where N is the number of waveguides, and becomes analogous to orbital angular momentum (OAM) degrees of freedom [8,9]. We have shown theoretically and experimentally that linear combinations of elastic states taking the form of tensor products of OAM and spinor amplitudes can form nonseparable states reminiscent of “entangled” Bell states [10]. We demonstrated that the amplitude coefficients of the nonseparable superposition of states are complex due to dissipation in the constitutive elastic materials. By tuning these complex amplitudes, we have shown that we can experimentally navigate a sizeable portion of the Bell state’s Hilbert space. These states lie in the tensor product Hilbert space of the subspaces associated with the direction of propagation and OAM degrees of freedom. The dimension of this product space scales linearly with the number of waveguides as 2 *N*. In order to achieve the full potential of the second quantum revolution, it would be highly desirable to construct classical nonseparable states that lie in an exponentially complex Hilbert space. More specifically, one would like to design a multipartite elastic system composed of *N* subsystems, each of which is able to be in at least two states. The dimension of the Hilbert space of such a system can take the value $2^{N}$, which scales exponentially with the number *N.* In this paper, we devise such a system. This system is composed of 1D elastic waveguides which are coupled via nonlinear forces. The nonlinear forces are quadratic functions of the relative elastic displacements in adjacent waveguides. The elastic medium is also assumed to be dissipative. The elastic nonlinearity by enabling wave–wave interaction is necessary to achieve wave mixing and, therefore, allowing the formation of waves for which the frequency and wave number are the sum of the frequencies and wave numbers of parent linear waves. These nonlinear waves are product waves of the parent waves and, therefore, span a Hilbert space, which is the product of the spaces supporting the parent waves. It is in that space which exploration is enabled by the nonlinearity that one can observe nonseparable superpositions of elastic waves. In this paper, the nonlinear coupling is treated as a perturbation. Here, we develop a multiple-time-scale perturbation theory for a system composed of three nonlinearly coupled waveguides driven by an external harmonic force. The driven elastic system can be treated as a multipartite system composed of subsystems (each subsystem corresponding to a well-defined OAM elastic band). Considering finite waveguides and, therefore, discrete plane wave states, for each OAM band, we limit the plane wave solutions to those two states that are in the near vicinity of the frequency of the external driver. We show that to first order in perturbation, if we excite two OAM bands and two plane wave states in each band, the elastic system can be visualized as a two-partite two-level system which can support superpositions of nonlinear modes which span an exponentially complex Hilbert space. This space, of dimension $2^{2}$, is the tensor product of the Hilbert space of two two-level subsystems. These nonlinear modes are shown to be nonseparable. The amplitudes of the superposition of nonlinear modes are complex due to dissipation and can be tuned by varying the driving frequency. Such behavior is a local analogue of the nonseparable superpositions of states of two-particle two-level quantum systems.

## 2. Model System and Method

We consider a system composed of three coupled one-dimensional mass-spring chains (Figure 1).

The chains and coupling springs obey linear elasticity. The discrete linear elastic equations of motion are given by the following:(1a)$$m\frac{\partial^{2}u_{n}}{\partial t^{2}}-k_{nn}\left(u_{n+1}-2u_{n}+u_{n-1}\right)-k_{c}\left(v_{n}-u_{n}\right)+\eta \frac{\partial u_{n}}{\partial t}=0$$
(1b)$$m\frac{\partial^{2}v_{n}}{\partial t^{2}}-k_{nn}\left(v_{n+1}-2v_{n}+v_{n-1}\right)-k_{c}\left(u_{n}-v_{n}\right)-k_{c}\left(w_{n}-v_{n}\right)+\eta \frac{\partial v_{n}}{\partial t}=0$$
(1c)$$m\frac{\partial^{2}w_{n}}{\partial t^{2}}-k_{nn}\left(w_{n+1}-2w_{n}+w_{n-1}\right)-k_{c}\left(v_{n}-w_{n}\right)+\eta \frac{\partial w_{n}}{\partial t}=0$$

In Equations (1a)–(1c), $u_{n},v_{n}$ and $w_{n}$ are the displacements of the $n^{th}$ mass of chain 1, 2, and 3, respectively. m is the mass, and the viscous damping coefficient η models the dissipation. $k_{nn}$ is the stiffness of the springs in the waveguide chains. Here, we take the coupling constant between chains $k_{c}$ to be the same for all coupled chains. In the limit of long wavelength compared to the inter-mass spacing, a, the equations of motions (1) of the three coupled linear harmonic chains of masses and springs become the following:(2a)$$\frac{\partial^{2}u}{\partial t^{2}}-\beta^{2}\frac{\partial^{2}u}{\partial x^{2}}-\alpha^{2}\left(v-u\right)+\overset{´}{\eta }\frac{\partial u}{\partial t}=0,$$
(2b)$$\frac{\partial^{2}v}{\partial t^{2}}-\beta^{2}\frac{\partial^{2}v}{\partial x^{2}}-\alpha^{2}\left(u-v\right)-\alpha^{2}\left(w-v\right)+\overset{´}{\eta }\frac{\partial v}{\partial t}=0,$$
(2c)$$\frac{\partial^{2}w}{\partial t^{2}}-\beta^{2}\frac{\partial^{2}w}{\partial x^{2}}-\alpha^{2}\left(v-w\right)+\overset{´}{\eta }\frac{\partial w}{\partial t}=0,$$
where $\beta^{2}=k_{nn}a^{2}/m$, $\alpha^{2}=k_{c}/m$ and $\overset{´}{\eta }=\eta /m$.

The coupling terms in Equations (2a)–(2c) can be grouped into α coupling matrix form $\left(\begin{array}{ccc} 1 & -1 & 0\\ -1 & 2 & -1\\ 0 & -1 & 1 \end{array}\right)$ acting on the vector $\left(\begin{array}{c} u\\ v\\ w \end{array}\right)$. The eigen vectors of the coupling matrix are isomorphic to an orbital angular momentum (OAM). The three normalized OAM eigen vectors corresponding to the eigen values $\lambda_{1}=0$, $\lambda_{1}=1$, and $\lambda_{3}=3$, are: $e_{1}=\left(\begin{array}{c} e_{1u}\\ e_{1v}\\ e_{1w} \end{array}\right)=\frac{1}{\sqrt{3}}\left(\begin{array}{c} 1\\ 1\\ 1 \end{array}\right),e_{2}=\left(\begin{array}{c} e_{2u}\\ e_{2v}\\ e_{2w} \end{array}\right)=\frac{1}{\sqrt{2}}\left(\begin{array}{c} 1\\ 0\\ -1 \end{array}\right),e_{3}=\left(\begin{array}{c} e_{3u}\\ e_{3v}\\ e_{3w} \end{array}\right)=\frac{1}{\sqrt{6}}\left(\begin{array}{c} 1\\ -2\\ 1 \end{array}\right).$ The associated dispersion relations for plane wave solutions, $e^{ikx}e^{i\omega_{k}t}$, are given by $\omega_{k}^{2}={\left(\beta k\right)}^{2}$, $\omega_{k}^{2}={\left(\beta k\right)}^{2}+{\left(\alpha \right)}^{2}$ and $\omega_{k}^{2}={\left(\beta k\right)}^{2}+3{\left(\alpha \right)}^{2}$.

The coupled elastic system is then driven externally with the external force ${\vec{F}}_{0}=\left(\begin{array}{c} F_{0}^{u}\\ F_{0}^{v}\\ F_{0}^{w} \end{array}\right)e^{i\omega t}=\vec{F}e^{i\omega t}$ applied at x=0.

The equations of motion of the driven coupled system become the following:(3a)$$\frac{\partial^{2}u}{\partial t^{2}}-\beta^{2}\frac{\partial^{2}u}{\partial x^{2}}-\alpha^{2}\left(v-u\right)+\overset{´}{\eta }\frac{\partial u}{\partial t}=F_{0}^{u}e^{i\omega t}\delta_{x=0},$$
(3b)$$\frac{\partial^{2}v}{\partial t^{2}}-\beta^{2}\frac{\partial^{2}v}{\partial x^{2}}-\alpha^{2}\left(u-v\right)-\alpha^{2}\left(w-v\right)+\overset{´}{\eta }\frac{\partial v}{\partial t}=F_{0}^{v}e^{i\omega t}\delta_{x=0},$$
(3c)$$\frac{\partial^{2}w}{\partial t^{2}}-\beta^{2}\frac{\partial^{2}w}{\partial x^{2}}-\alpha^{2}\left(v-w\right)+\overset{´}{\eta }\frac{\partial w}{\partial t}=F_{0}^{w}e^{i\omega t}\delta_{x=0},$$

At a steady state, the displacement field takes the form of a linear combination of frequency modes:(4)$$\left(\begin{array}{c} u\left(x,t\right)\\ v\left(x,t\right)\\ w\left(x,t\right) \end{array}\right)=\left[\underset{k_{1}}{\sum }A_{1}\left(k_{1}\right)e_{1}e^{ik_{1}x}+\underset{k_{2}}{\sum }A_{2}\left(k_{2}\right)e_{2}e^{ik_{2}x}+\underset{k_{3}}{\sum }A_{3}\left(k_{3}\right)e_{3}e^{ik_{3}x}\right]e^{i\omega t}$$
where
$$A_{1}\left(k\right)=\frac{\frac{1}{\sqrt{3}}\left(\begin{array}{c} 1\\ 1\\ 1 \end{array}\right).\vec{F}}{\omega_{01}^{2}\left(k\right)-\omega^{2}-i\overset{´}{\eta }\omega },A_{2}\left(k\right)=\frac{\frac{1}{\sqrt{2}}\left(\begin{array}{c} 1\\ 0\\ -1 \end{array}\right).\vec{F}}{\omega_{02}^{2}\left(k\right)-\omega^{2}-i\overset{´}{\eta }\omega },A_{3}\left(k\right)=\frac{\frac{1}{\sqrt{6}}\left(\begin{array}{c} 1\\ -2\\ 1 \end{array}\right).\vec{F}}{\omega_{03}^{2}\left(k\right)-\omega^{2}-i\overset{´}{\eta }\omega },$$
and writing $A_{1}\left(k\right)=\left|A_{1}\left(k\right)\right|e^{i\phi_{1}\left(k\right)}$, $A_{2}\left(k\right)=\left|A_{2}\left(k\right)\right|e^{i\phi_{2}\left(k\right)}$, and $A_{3}\left(k\right)=\left|A_{3}\left(k\right)\right|e^{i\phi_{3}\left(k\right)}$, we find $\phi_{1}\left(k\right)=tan^{1}\left(\frac{\overset{´}{\eta }\omega }{\omega_{01}^{2}\left(k\right)-\omega^{2}}\right)$, $\phi_{2}\left(k\right)=tan^{1}\left(\frac{\overset{´}{\eta }\omega }{\omega_{02}^{2}\left(k\right)-\omega^{2}}\right)$ and $\phi_{3}\left(k\right)=tan^{1}\left(\frac{\overset{´}{\eta }\omega }{\omega_{03}^{2}\left(k\right)-\omega^{2}}\right)$. Here, $\omega_{01},\omega_{02}$ and $\omega_{03}$ are the eigen frequencies of the (1 1 1), (1 0 −1) and (1 −2 1) OAM eigen vectors. Equations (3a)–(3c) relate to infinite chains, however, in the case of more realistic finite spring-mass chains, one expects to deal with a finite set of modes labeled by a discrete set of wavenumbers. In Equation (4), we have used discrete summation as a proxy for a finite system. In that case and in light of the Lorentzian line shape of the amplitudes, $A_{1,2,3}$, it is possible to conceive the use of isofrequency drivers which minimize the amplitude of the $e_{1}$ OAM eigen modes compared to that of $e_{2}$ and $e_{3}$. For instance, exploiting the orthogonality of $e_{1}$, $e_{2}$ and $e_{3}$, one may employ a driving force $\vec{F}$, which is a linear combination of $e_{2}$ and $e_{3}$. In that case, the first summation in Equation (4) will have only negligible amplitudes. We subsequently limit the second summation to the two states $k_{2}$ and $k_{2}^{\prime }$ with OAM eigen vector $e_{2}$ and frequency ω that contribute the largest amplitudes, namely $A_{2}=A_{2}\left(k_{2}\right)$ and $A_{2}^{\prime }=A_{2}\left(k_{2}^{\prime }\right)$. Similarly, the third sum in Equation (4) is also limited to the two states $k_{3}$ and $k_{3}^{\prime }$ with the largest amplitudes $A_{3}=A_{3}\left(k_{3}\right)$ and $A_{3}^{\prime }=A_{3}\left(k_{3}^{\prime }\right)$. These states are illustrated in Figure 2. Note that here, for the sake of simplicity of our demonstration, we use only positive wavenumbers but finite mass-spring chains would also support negative wavenumber states.

Equation (4) reduces then to the following:(5)$$\left(\begin{array}{c} u\left(x,t\right)\\ v\left(x,t\right)\\ w\left(x,t\right) \end{array}\right)=\left\{\left[A_{2}e^{ik_{2}x}+A_{2}^{\prime }e^{ik_{2}^{\prime }x}\right]e_{2}+\left[A_{3}e^{ik_{3}x}+A_{3}^{\prime }e^{ik_{3}^{\prime }x}\right]e_{3}\right\}e^{i\omega t}$$

Equation (5) represents the superposition of states of a two-partite system (with subsystems identified by their OAM index, 2 and 3) possessing two levels (i.e., primed and unprimed wave numbers). The amplitudes in this superposition are complex due to their Lorentzian character. This means that one can tune the relative phase associated with these amplitudes by controlling the frequency ω. $e^{ik_{2}x}$ and $e^{ik_{2}^{\prime }x}$ form a basis for subsystem 2. $e^{ik_{3}x}$ and $e^{ik_{3}^{\prime }x}$ form a basis for subsystem 3.

We now consider the coupled system with nonlinear coupling springs. The equations of motion in the long wavelength limit take the form:(6a)$$\frac{\partial^{2}u}{\partial t^{2}}-\beta^{2}\frac{\partial^{2}u}{\partial x^{2}}-\alpha^{2}\left(v-u\right)+\varepsilon {\left(v-u\right)}^{2}+\overset{´}{\eta }\frac{\partial u}{\partial t}=F_{0}^{u}e^{i\omega t}\delta_{x=0}$$
(6b)$$\frac{\partial^{2}v}{\partial t^{2}}-\beta^{2}\frac{\partial^{2}v}{\partial x^{2}}-\alpha^{2}\left(u-v\right)-\alpha^{2}\left(w-v\right)+\varepsilon {\left(u-v\right)}^{2}+\epsilon {\left(w-v\right)}^{2}+\overset{´}{\eta }\frac{\partial v}{\partial t}=F_{0}^{v}e^{i\omega t}\delta_{x=0}$$
(6c)$$\frac{\partial^{2}w}{\partial t^{2}}-\beta^{2}\frac{\partial^{2}w}{\partial x^{2}}-\alpha^{2}\left(v-w\right)+\varepsilon {\left(v-w\right)}^{2}+\overset{´}{\eta }\frac{\partial w}{\partial t}=F_{0}^{w}e^{i\omega t}\delta_{x=0}$$

The nonlinear quadratic terms correspond to forces that do not depend on the sign of the relative displacements. This type of nonlinearity will occur in heterogeneous materials that contain microcracks, for example [11]. This type of nonlinear term will lead to states with doubled frequency.

We now attempt to solve Equations (6a)–(6c) using multiple-time-scale perturbation theory [12]; the nonlinear term being the perturbation. For this, we consider ε to be a small quantity. The advantage of multiple-time-scale perturbation theory is that it can capture nonlinear amplitude–frequency interaction in systems with time-dependent amplitude and phase [13]. This is the case when multiple waves with comparable amplitudes interact with each other, leading to multiple secular terms in the dynamical equations. The additional degrees of freedom introduced via the multiple time scales are necessary to obtain a perturbation solution. While approaches such as the molecular dynamics method can also be utilized [14] to numerically solve nonlinear dynamical problems, analytical methods are still useful tools to illuminate the multiple wave scattering processes in nonlinear one-dimensional mass-spring chains, such as the one studied here.

We rewrite the displacements as polynomials in ε, that is, considering, as an example, the displacement u:(7)$$u\left(\tau_{0},\tau_{1},\tau_{2}\right)=u_{0}\left(\tau_{0},\tau_{1},\tau_{2}\right)+\varepsilon u_{1}\left(\tau_{0},\tau_{1},\tau_{2}\right)+\varepsilon^{2}u_{2}\left(\tau_{0},\tau_{1},\tau_{2}\right)+\ldots$$
with $\tau_{0}=t$, $\tau_{1}=\varepsilon t$, $\tau_{2}=\varepsilon^{2}t.$

We also have the following:(8)$$\frac{\partial u}{\partial t}=\frac{\partial u_{0}}{\partial \tau_{0}}+\varepsilon \left(\frac{\partial u_{1}}{\partial \tau_{0}}+\frac{\partial u_{0}}{\partial \tau_{1}}\right)+\varepsilon^{2}\left(\frac{\partial u_{2}}{\partial \tau_{0}}+\frac{\partial u_{1}}{\partial \tau_{1}}+\frac{\partial u_{0}}{\partial \tau_{2}}\right)+\ldots$$
and
(9)$$\frac{\partial^{2}u}{\partial t^{2}}=\frac{\partial^{2}u_{0}}{\partial \tau_{0}^{2}}+\varepsilon \left(\frac{\partial^{2}u_{1}}{\partial \tau_{0}^{2}}+2\frac{\partial^{2}u_{0}}{\partial \tau_{1}\partial \tau_{0}}\right)+\varepsilon^{2}\left(\frac{\partial^{2}u_{2}}{\partial \tau_{0}^{2}}+2\frac{\partial^{2}u_{1}}{\partial \tau_{1}\partial \tau_{0}}+2\frac{\partial^{2}u_{0}}{\partial \tau_{2}\partial \tau_{0}}+\frac{\partial^{2}u_{0}}{\partial \tau_{1}^{2}}\right)+\ldots$$

Similar expressions can be obtained for the other displacements, v and w.

We illustrate below the expansion to first order of the nonlinear terms in Equation (6a):(10)$${\left(v-u\right)}^{2}={\left(v_{0}-u_{0}\right)}^{2}+\varepsilon \left[\left(v_{0}-u_{0}\right)\left(v_{1}-u_{1}\right)+\left(v_{1}-u_{1}\right)\left(v_{0}-u_{0}\right)\right]+\ldots$$

Inserting Expressions (7)–(10) and their equivalent forms for the other displacements into Equations (6a)–(6c) leads to one equation to zeroth order in the perturbation, ε, one equation to first order in ε and equations for subsequent higher orders. Here, we limit ourselves to the zeroth and first-order equations in the perturbation. To zeroth order, we obtain the following:(11a)$$\frac{\partial^{2}u_{0}}{\partial \tau_{0}^{2}}-\beta^{2}\frac{\partial^{2}u_{0}}{\partial x^{2}}-\alpha^{2}\left(v_{0}-u_{0}\right)+\overset{´}{\eta }\frac{\partial u_{0}}{\partial \tau_{0}}=F_{0}^{u}e^{i\omega \tau_{0}}\delta_{x=0},$$
(11b)$$\frac{\partial^{2}v_{0}}{\partial \tau_{0}^{2}}-\beta^{2}\frac{\partial^{2}v_{0}}{\partial x^{2}}-\alpha^{2}\left(u_{0}-v_{0}\right)-\alpha^{2}\left(w_{0}-v_{0}\right)+\overset{´}{\eta }\frac{\partial v}{\partial \tau_{0}}=F_{0}^{v}e^{i\omega \tau_{0}}\delta_{x=0},$$
(11c)$$\frac{\partial^{2}w_{0}}{\partial \tau_{0}^{2}}-\beta^{2}\frac{\partial^{2}w_{0}}{\partial x^{2}}-\alpha^{2}\left(v_{0}-w_{0}\right)+\overset{´}{\eta }\frac{\partial w_{0}}{\partial \tau_{0}}=F_{0}^{w}e^{i\omega \tau_{0}}\delta_{x=0},$$

This is essentially, Equations (3a)–(3c) with already known solutions.

Regrouping all terms multiplied by ε, we obtain the first-order equations:(12a)$$\frac{\partial^{2}u_{1}}{\partial \tau_{0}^{2}}+2\frac{\partial^{2}u_{0}}{\partial \tau_{1}\partial \tau_{0}}-\beta^{2}\frac{\partial^{2}u_{1}}{\partial x^{2}}-\alpha^{2}\left(v_{1}-u_{1}\right)+\overset{´}{\eta }\left(\frac{\partial u_{1}}{\partial \tau_{0}}+\frac{\partial u_{0}}{\partial \tau_{1}}\right)=-{\left(v_{0}-u_{0}\right)}^{2},$$
(12b)$$\frac{\partial^{2}v_{1}}{\partial \tau_{0}^{2}}+2\frac{\partial^{2}v_{0}}{\partial \tau_{1}\partial \tau_{0}}-\beta^{2}\frac{\partial^{2}v_{1}}{\partial x^{2}}-\alpha^{2}\left(u_{1}-v_{1}\right)-\alpha^{2}\left(w_{1}-v_{1}\right)+\overset{´}{\eta }\left(\frac{\partial v_{1}}{\partial \tau_{0}}+\frac{\partial v_{0}}{\partial \tau_{1}}\right)\;=-{\left(u_{0}-v_{0}\right)}^{2}-{\left(w_{0}-v_{0}\right)}^{2},$$
(12c)$$\frac{\partial^{2}w_{1}}{\partial \tau_{0}^{2}}+2\frac{\partial^{2}w_{0}}{\partial \tau_{1}\partial \tau_{0}}-\beta^{2}\frac{\partial^{2}w_{1}}{\partial x^{2}}-\alpha^{2}\left(v_{1}-w_{1}\right)+\overset{´}{\eta }\left(\frac{\partial w_{1}}{\partial \tau_{0}}+\frac{\partial w_{0}}{\partial \tau_{1}}\right)=-{\left(v_{0}-w_{0}\right)}^{2},$$

We note that while the external force $\vec{F}$ drives the zeroth-order equation, only the zeroth-order solutions drive the first-order equations. Note that for the purpose of this investigation, we do not need to expand the equations of motion to second order in perturbation. Expansion to that order would provide, in addition to obtaining the second-order solutions, corrections to the zeroth-order solutions. The solutions of Equations (12a)–(12c) are sums of the solutions of the homogeneous equations (i.e., without the terms on the right-hand side of the equals sign) and particular solutions (i.e., with the right-hand-side terms). In order to eliminate secular terms in the homogeneous solutions, we impose to $u_{0}$, $v_{0}$, $w_{0}$, not to be functions of $\tau_{1}$. In that case, the derivatives $\frac{\partial^{2}}{\partial \tau_{1}\partial \tau_{0}}$ and $\frac{\partial }{\partial \tau_{1}}$ in Equations (12a)–(12c) are effectively zero.

We use Equation (5) to write the solution to the zeroth-order equations:(13)$$\left(\begin{array}{c} u_{0}\left(x,\tau_{0}\right)\\ v_{0}\left(x,\tau_{0}\right)\\ w_{0}\left(x,\tau_{0}\right) \end{array}\right)=\left\{\left[A_{2}e^{ik_{2}x}+A_{2}^{\prime }e^{ik_{2}^{\prime }x}\right]e_{2}+\left[A_{3}e^{ik_{3}x}+A_{3}^{\prime }e^{ik_{3}^{\prime }x}\right]e_{3}\right\}e^{i\omega \tau_{0}}$$

For the sake of illustration, we derive the zeroth-order driving term in Equation (11a), namely:
(14)$$\begin{array}{ll} {\left(v_{0}-u_{0}\right)}^{2}= & {\left[A_{2}e^{ik_{2}x}+A_{2}^{\prime }e^{ik_{2}^{\prime }x}\right]}^{2}{\left(e_{2v}-e_{2u}\right)}^{2}e^{i2\omega \tau_{0}}+{\left[A_{3}e^{ik_{3}x}+A_{3}^{\prime }e^{ik_{3}^{\prime }x}\right]}^{2}{\left(e_{3v}-e_{3u}\right)}^{2}e^{i2\omega \tau_{0}}\\ & \;+2[A_{2}A_{3}e^{ik_{2}x}e^{ik_{3}x}+A_{2}A_{3}^{\prime }e^{ik_{2}x}e^{ik_{3}^{\prime }x}+A_{2}^{\prime }A_{3}e^{ik_{2}^{\prime }x}e^{ik_{3}x}\\ & \;+A_{2}^{\prime }A_{3}^{\prime }e^{ik_{2}^{\prime }x}e^{ik_{3}^{\prime }x}]e^{i2\omega \tau_{0}}\left(e_{2v}-e_{2u}\right)\left(e_{3v}-e_{3u}\right) \end{array}$$

The first term and second term lead to self-interaction between the two states of the same subsystems. The third term containing cross-terms between states of different subsystems (i.e., different OAM) enables us to explore the tensor product Hilbert space of the Hilbert spaces of the subsystems. Let us consider the two-dimensional Hilbert space of the subsystem 2,
$H_{2}$, with basis $e^{ik_{2}x}$ and $e^{ik_{2}^{\prime }x}$ and the two-dimensional Hilbert space of subsystem 3, $H_{3}$, with basis $e^{ik_{3}x}$ and $e^{ik_{3}^{\prime }x}$. The tensor product space, $H_{23}=H_{2}⊗H_{3}$ has dimension $2^{2}$ with the basis $\varphi_{1}=e^{i\left(k_{2}+k_{3}\right)x}$, $\varphi_{2}=e^{i\left(k_{2}+k_{3}^{\prime }\right)x}$, $\varphi_{3}=e^{i\left(k_{2}^{\prime }+k_{3}\right)x}$, $\varphi_{4}=e^{i\left(k_{2}^{\prime }+k_{3}^{\prime }\right)x}$. It is these cross-terms which enable us to use nonlinearity to explore the tensor product space of the bipartite elastic system. We will now focus on the cross-terms as driving terms for the first-order equation. After deriving expressions for the other zeroth order cross-terms of, ${\left(u_{0}-v_{0}\right)}^{2}$, ${\left(w_{0}-v_{0}\right)}^{2}$ and ${\left(v_{0}-w_{0}\right)}^{2}$ and inserting into Equations (12a)–(12c), we seek the cross-terms contributions to the particular solutions of the first-order equations in the form:(15a)$$u_{1}=\left(a_{u}\varphi_{1}+b_{u}\varphi_{2}+c_{u}\varphi_{3}+d_{u}\varphi_{4}\right)e^{i2\omega \tau_{0}}$$
(15b)$$v_{1}=\left(a_{v}\varphi_{1}+b_{v}\varphi_{2}+c_{v}\varphi_{3}+d_{v}\varphi_{4}\right)e^{i2\omega \tau_{0}}$$
(15c)$$w_{1}=\left(a_{w}\varphi_{1}+b_{w}\varphi_{2}+c_{w}\varphi_{3}+d_{w}\varphi_{4}\right)e^{i2\omega \tau_{0}}$$

We seek expressions for the 12 coefficients $a_{u}$, $b_{u}$, …, $c_{w},$ and $d_{w}$. Using the definition of the OAM eigen vectors, after extensive algebraic manipulations, the contribution of zeroth-order cross-terms to the first-order particular solutions becomes the following:(16)$$\left(\begin{array}{c} u_{1}\\ v_{1}\\ w_{1} \end{array}\right)=\sqrt{3}\left(\begin{array}{c} 1\\ 0\\ -1 \end{array}\right)\left[\frac{A_{2}A_{3}}{F_{1}}\varphi_{1}+\frac{A_{2}A_{3}^{\prime }}{F_{2}}\varphi_{2}+\frac{A_{2}^{\prime }A_{3}}{F_{3}}\varphi_{3}+\frac{A_{2}^{\prime }A_{3}^{\prime }}{F_{4}}\varphi_{4}\right]e^{i2\omega \tau_{0}}$$
where
(17a)$$F_{1}=-4\omega^{2}+\beta^{2}{\left(k_{2}+k_{3}\right)}^{2}+\alpha^{2}+i2\overset{´}{\eta }\omega ,$$
(17b)$$F_{2}=-4\omega^{2}+\beta^{2}{\left(k_{2}+k_{3}^{\prime }\right)}^{2}+\alpha^{2}+i2\overset{´}{\eta }\omega ,$$
(17c)$$F_{3}=-4\omega^{2}+\beta^{2}{\left(k_{2}^{\prime }+k_{3}\right)}^{2}+\alpha^{2}+i2\overset{´}{\eta }\omega ,$$
(17d)$$F_{4}=-4\omega^{2}+\beta^{2}{\left(k_{2}^{\prime }+k_{3}^{\prime }\right)}^{2}+\alpha^{2}+i2\bar{\eta }\omega .$$

It is worth noting that the $F_{I}$, *I* = 1, 2, 3, 4 are complex quantities due to the dissipation. The complex nature of these amplitudes completes the analogy between superpositions of elastic waves and superposition of states of a two-partite two-level quantum system.

Equation (16) represents a state of the nonlinear system in the tensor-product Hilbert space $H_{23}$. This state is said to be separable if it can be written as a tensor product of individual states of the two subsystems. This condition is satisfied if we can find four complex numbers $\rho_{2}$, $\rho_{2}^{\prime }$, $\rho_{3}$, $\rho_{3}^{\prime }$ such that
(18)$$\begin{array}{l} \left[\frac{A_{2}A_{3}}{F_{1}}\varphi_{1}+\frac{A_{2}A_{3}^{\prime }}{F_{2}}\varphi_{2}+\frac{A_{2}^{\prime }A_{3}}{F_{3}}\varphi_{3}+\frac{A_{2}^{\prime }A_{3}^{\prime }}{F_{4}}\varphi_{4}\right]e^{i2\omega \tau_{0}}\\ \;=\left[\frac{A_{2}}{\rho_{2}}e^{ik_{2}x}+\frac{A_{2}^{\prime }}{\rho_{2}^{\prime }}e^{ik_{2}^{\prime }x}\right]e^{i\omega \tau_{0}}\left[\frac{A_{3}}{\rho_{3}}e^{ik_{3}x}+\frac{A_{3}^{\prime }}{\rho_{3}^{\prime }}e^{ik_{3}^{\prime }x}\right]e^{i\omega \tau_{0}} \end{array}$$

This condition reduced to the factorization of the $F_{I}$’s in the form
$$F_{1}=\rho_{2}\rho_{3}\;F_{2}=\rho_{2}\rho_{3}^{\prime }\;F_{3}=\rho_{2}^{\prime }\rho_{3}\;F_{4}=\rho_{2}^{\prime }\rho_{3}^{\prime }$$

If we define the complex numbers $\rho_{2}=X_{2}+iY_{2}$, $\rho_{3}=X_{3}+iY_{3}$ and $\rho_{3}^{\prime }=X_{3}^{\prime }+iY_{3}^{\prime }$, the first two conditions for factorization imply for the imaginary part of $F_{1}$ and $F_{2}$ that $2\overset{´}{\eta }\omega =X_{2}Y_{3}+Y_{2}X_{3}=X_{2}Y_{3}^{\prime }+Y_{2}X_{3}^{\prime }$. This first equation can only be satisfied if $\rho_{3}^{\prime }=\rho_{3}$. Furthermore, the real parts of $F_{1}$ and $F_{2}$ take the form $-4\omega^{2}+\beta^{2}{\left(k_{2}+k_{3}\right)}^{2}+\alpha^{2}=X_{2}X_{3}-Y_{2}Y_{3}$ and $-4\omega^{2}+\beta^{2}{\left(k_{2}+k_{3}^{\prime }\right)}^{2}+\alpha^{2}=X_{2}X_{3}^{\prime }-Y_{2}Y_{3}^{\prime }$. In that case, the real parts of $F_{1}$ and $F_{2}$ need to be the same, that is, one needs to impose the impossible condition $k_{3}=k_{3}^{\prime }$. We, therefore, have proven that the particular solutions of the first-order equation resulting from the nonlinearity (i.e., cross zeroth-order driving terms) given by Equation (16) which resides in the tensor-product Hilbert space $H_{23}=H_{2}⊗H_{3}$ is not separable, i.e., it is not factorizable into the product of a solution supported by the Hilbert space $H_{2}$ and a solution supported by $H_{3}$.

These nonseparable states are defined in the two-partite Hilbert space $H_{23}$, for which the dimension $2^{2}$ is exponentially complex. Accessibility to larger multipartite Hilbert spaces would require elastic systems composed of N mass-spring chains with nonlinear coupling scaling as a power of N. Provided that each one of the N OAM bands can be treated as a two-level subsystem, the multipartite nonlinear system would admit elastic states to first order that span the exponentially complex Hilbert space of dimension $2^{N}$. This approach can be generalized to subsystems with more than two levels. Indeed, the external harmonic driver may excite more than two discrete states along a given OAM band. In that case, denoting by B the number of such plane wave states, i.e., the dimension of the base for the driven elastic states, and by E the exponent of the nonlinear coupling, the first-order nonlinear states will span a product space of dimension $B^{E}$. In the event that E<N, one may not achieve complete exponential complexity but superlinear complexity

## 3. Conclusions

We have used multiple-time-scale perturbation theory, to investigate the behavior of an externally driven elastic system composed of three coupled mass-spring chains. The chains are coupled via nonlinear springs whose force depends quadratically on the relative displacement of adjacent chains. The quadratic nonlinear forces in Equations (6a)–(6c) provide a path toward realizing nonseparable, or in other words, “classically entangled” elastic states. The possibility of realizing physically with elastic waves nonseparable superpositions of states with superlinear complexity if not exponential complexity opens new doors in the area of elastic-wave-supported information processing. “Classically entangled” elastic states are local. However, nonlocality is not a necessary condition for applying the concept of nonseparability to processing information. Nonseparable superpositions of elastic waves offer the advantage of stability over entangled states of true quantum systems. Nonseparable superpositions of elastic waves are robust against decoherence and will not require operating at cryogenic temperatures to maintain the delicate balance of the superpositions. Nonseparable superpositions of elastic waves do not suffer from the phenomenon of wave function collapse upon measurement. A coherent superposition of quantum states collapses into a pure state upon measurement. Multiple statistical measurements are, therefore, necessary to obtain information on the original superposition. From a physical point of view, following work reported in [10], the realization of the coupled systems described here theoretically could be achieved by using finite-length cylindrical elastic rods as the waveguides and employ a nonlinear elastic material as the coupling agent between the rods. Future research would involve experimental measurement of the exponentially complex nonseparable superpositions of elastic waves in these nonlinear arrays of coupled waveguides. The OAM states could be directly excited using transducer technologies while the spatial (wave number) characteristics would be subsequently measurable using noncontact methods such as scanning laser Doppler vibrometry. This future work would demonstrate the scalability of the superpositions introduced in this paper.

## Figures and Tables

**Figure 1 materials-12-03553-f001:**
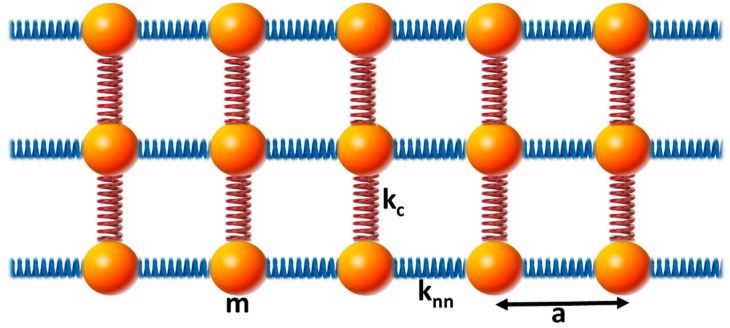
Schematic illustration of the system composed of three coupled one-dimensional elastic waveguides. The red springs couple the three elastic chains identified with blue springs. See text for meaning of symbols.

**Figure 2 materials-12-03553-f002:**
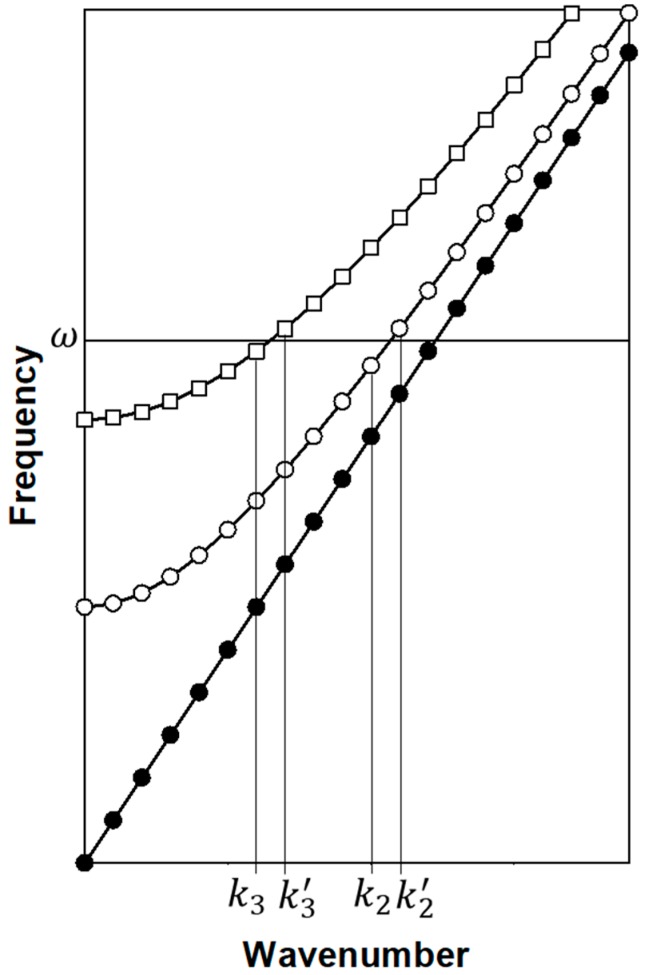
Schematic illustration of the band structure of the three coupled one-dimensional elastic waveguides. ω is the frequency of the external driving force. The wavenumbers of the states with the largest amplitudes along the two bands with cut-off frequencies are labelled on the wavenumber axis.

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
