# Peer review of "Exponentially Complex “Classically Entangled” States in Arrays of One-Dimensional Nonlinear Elastic Waveguides"

_materials, 2019, doi:10.3390/ma12213553_

Round 1
Reviewer 1 Report
Review of the manuscript entitled:
Exponentially complex "classically entangled" states in arrays of one-dimensional nonlinear elastic waveguides
The submitted manuscript proposes novel classical counterparts of quantum entangled states implemented within the realm of classical theory of elasticity, employing one dimensional elastic waveguides coupled via elastic nonlinearity of the surrounding medium. The research presented in the manuscript may lead to development of classical analogues to quantum computers with potential applications in cryptography, telecommunication, sensors, etc. Indeed, quantum computers, based on "pure" quantum effects still struggle to prove their superiority over the conventional computers, despite two decades of extensive R&D efforts. "Classical" quantum computers based on physical phenomena described in the domain of classical physics, such as theory of elasticity or electromagnetism, may bring breakthrough in practical advances of the second quantum revolution. In fact, nonseparable superpositions of elastic waves (classical entanglement) are robust, not prone to decoherence and do not require cryogenic temperatures.
The quality of the submitted manuscript will be improved if the authors take into account the following recommendations:
The authors should explain why they use elastic nonlinearity to achieve local nonseparable superpositions of elastic waves not other properties of elastic materials, such as elasticity, viscosity, etc. Is nonlinearity the only way to achieve this goal?
The authors should explain why they employed a multiple time scale perturbation technique in their calculations? Are there any other techniques, including numerical methods, which could be used to this end?
In the second sentence below the formula 6.c the authors write:
This type of nonlinearity will occur in heterogeneous materials that contain microcracks
This statement should be supported by a relevant reference.
The authors should briefly state in Conclusions possibilities of future research following the results obtained in the submitted manuscript.
The submitted manuscript is a valuable contribution to the second quantum revolution implemented in the realm of the classical physics and should be published after implementing 4 recommendations, presented in this review.
The manuscript needs a minor revision.
Reviewer 2 Report
This paper investigated in a theoretical study to describe the behavior of an externally driven elastic system composed of three coupled mass-spring chains using multiple time scale perturbation theory which was induced by the nonlinear coupling. Thus, they first applied linear elasticity and then discussed the nonlinear coupling strings to imitate a two-partite two-level quantum system. This paper is interesting and well organized in a logical way, although they make several simplifications while building their model.
The conclusion part contains too many contents. The authors could divide it into discussion and conclusion parts. Only the main point are included in conclusion part.
